# Dissecting the NK Cell Population in Hematological Cancers Confirms the Presence of Tumor Cells and Their Impact on NK Population Function

**DOI:** 10.3390/vaccines8040727

**Published:** 2020-12-02

**Authors:** Dang-Nghiem Vo, Michael Constantinides, Nerea Allende-Vega, Catherine Alexia, Guillaume Cartron, Martin Villalba

**Affiliations:** 1IRMB, University Montpellier, INSERM, 34295 Montpellier, France; dang.nghiem_vo@med.lu.se (D.-N.V.); m-constantinides@chu-montpellier.fr (M.C.); nerea.allende-vega@inserm.fr (N.A.-V.); catherine.alexia@inserm.fr (C.A.); 2IRMB, CHU Montpellier, 34295 Montpellier, France; 3Département d’Hématologie Clinique, CHU Montpellier, 34295 Montpellier, France; g-cartron@chu-montpellier.fr; 4IRMB, University Montpellier, INSERM, CNRS, CHU Montpellier, 34295 Montpellier, France

**Keywords:** NK cells, UMAP, trogocytosis, AML, Hodgkin lymphoma

## Abstract

The lymphocyte lineage natural killer (NK) cell is part of the innate immune system and protects against pathogens and tumor cells. NK cells are the main cell effectors of the monoclonal antibodies (mAbs) that mediates antibody-dependent cell cytotoxicity (ADCC). Hence, it is relevant to understand NK physiology and status to investigate the biological effect of mAbs in the clinic. NK cells are heterogeneous with multiple subsets that may have specific activity against different attacks. The presence of viral-sculpted NK cell populations has already been described, but the presence of cancer-sculpted NK cells remains unknown. Cancer induces a broad NK cell dysfunction, which has not been linked to a specific population. Here, we investigated the NK cell population by Uniform Manifold Approximation and Projection (UMAP) embed maps in Hodgkin lymphoma (HL) and acute myeloid leukemia (AML) patients at diagnosis and at least 30 days after treatment, which correlates with tumor cell clearance. We found that the NK lineage largely responded to the tumor by generating antitumor NK cells and renewing the population with a subset of immature NK cells. However, we failed to identify a specific “memory-like” subset with the NK cell markers used. Moreover, in patients in relapse, we found essentially the same NK populations as those found at diagnosis, suggesting that NK cells equally respond to the first or second tumor rise. Finally, we observed that previous cytomegalovirus (CMV) infection largely affects the tumor-associated changes in NK population, but the CMV-associated CD57^+^NKG2C^+^ NK cell population does not appear to play any role in tumor immunity.

## 1. Introduction

The innate lymphocyte lineage natural killer (NK) cell protects against multiple pathogens and tumor cells [1]. The NK cell compartment is very heterogenous [2,3] and contains more subsets than originally thought [1]. An individual may have more than 30,000 distinct NK cell phenotypes in blood [2]. Some of them are “natural,” as they are present in multiple healthy individuals. In contrast, others could have been induced by environmental factors, such as viral infection, and other pathological conditions, such as the tumor microenvironment [4]. Identification of new subsets that play, or can play, a significant role in pathological conditions is relevant to better understand NK cell physiology. There is an antitumor subset that has been described in several hematological cancers, including acute myeloid leukemia (AML). This subset is characterized by expression of CD45RO (CD45RO cells) mostly together with CD45RA (CD45RARO cells). CD45RARO cells contain the cells that have degranulated and performed trogocytosis of tumor-expressed antigens [5,6,7]. Trogocytosis is a process via which NK cells acquire antigens from targets through membrane exchange, implying a close contact with tumor cells. This, together with the fact that these NK cells have degranulated, strongly suggests that these cells represent an anti-tumor population. However, it is unknown if these antitumor NK cells have effectively killed cancer cells. At the time of diagnosis, most hematological cancer patients, including acute myeloid leukemia (AML) patients, showed high levels of these CD45RARO cells. This increase was paralleled by a decrease in the CD45RA^+^RO^−^ population (CD45RA cells) [5,6,7]. After treatment and tumor clearing, there was a marked decrease in the percentage of CD45RARO, which is logically associated to a decrease of NK cells that had performed trogocytosis or degranulated. The presence of this antitumor NK population has not been studied in Hodgkin lymphoma (HL) patients.

The impact of disease in NK subsets is not exclusive of cancer. Other diseases including infection and autoimmunity induce impaired NK function and the emergence of NK subsets rarely observed in healthy individuals [8]. However, little is known about the fate of the disease-specific subsets. One exception is virus-associated memory, or adaptive, NK cells. For example, human cytomegalovirus (HCMV) infection promotes expansion of NKG2C^+^ NK cells with memory-like properties and the presence of NKG2C^+^CD57^+^ NK cells is associated with prior cytomegalovirus (CMV) infection [9,10]. However, it is unknown if these NK cells have a specific activity against HCMV itself or if they are specifically activated after HCMV reactivation [11]. Previous viral infection can affect the NK response to a second danger such as tumor growth. For example, most glioblastoma express CMV proteins, and CMV infection can imprint NK cells, perhaps because CMV alters the distribution of KIR–HLA ligand interactions and this modulates NK cell function [12]. In addition, KIR2DS2+ and KIR2DS4+ NK cells are more potent killers against glioblastoma cells than NK cells negative for those receptors [13]. Furthermore, the presence of the KIR allele KIR2DS4*00101 is prognostic of prolonged survival in glioblastoma patients [12]. During the acute phase of Epstein Barr virus (EBV) infection, there is accumulation of a proliferative NKG2A^+^KIR^−^CD57^−^ NK cell subset that remains elevated up to 6 months after the acute symptomatic phase [14]. Hence, two different types of herpesviruses, i.e., HCMV and EBV, promote expansion of different NK cell subsets. Another example of long-term atypical NK cell distribution is found in HIV-infected humans with the expansion of anergic CD56^−^/CD16^+^ NK cells [15].

In cancer patients, the phenotype of the associated NK cell subsets may vary according to the tumor and its anatomical location [16,17]. The continuous presence of tumor cells impairs NK cell function [18,19,20], which, in turn, helps tumor immune escape [21,22]. However, the persistence of specific subsets linked to the disease has not been shown yet.

Whereas NK cells are considered part of the innate immune system, they are lymphocytes, an immune population that contains the effector adaptive immune cells, i.e., T and B cells. These lineages rapidly react to the presence of targets. NK cells can indirectly mediate the adaptive immune response because they mediate antibody-dependent cell cytotoxicity (ADCC) and are, consequently, mediators of the humoral response. In the clinic, NK cells are getting increased attention because they are the main effectors of monoclonal antibodies (mAbs) inducing ADCC. Hence, to understand if a mAb would work in vivo is important to know the status of the effector NK cells. NK cells can mediate adaptive-like immune responses in humanized mice [23], but how the whole NK population adapts to the chronic presence of targets such as tumor cells in humans is unknown. We investigated it in HL and AML patients by using multiparameter flow cytometry and processed data by Uniform Manifold Approximation and Projection (UMAP) maps [24]. We found that NK lineage largely reacts to the chronic presence of targets with totally new populations appearing. In addition, we show that disease-related subsets profoundly evolve when targets disappear. This suggests that these populations depend on the presence of targets, and the presence of long-lasting populations is unclear.

## 2. Materials, Subjects, and Methods

### 2.1. Ethical Statement

The use of human specimens for scientific purposes was approved by the French National Ethics Committee. All methods were carried out in accordance with the approved guidelines and regulations of this committee. Written informed consent was obtained from each patient or donor prior to sampling.

### 2.2. HL and AML Patients

Data and samples from patients were collected at the Clinical Hematology Department of the CHU Montpellier, France, after patient’s written consent and following French regulations. Patients were enrolled in the HEMODIAG_2020 (ID-RCB: 2011-A00924-37) clinical program approved by the “Comités de Protection des Personnes Sud Méditerranée I” with the reference 1324. Samples were collected at diagnosis and kept by the CHU Montpellier [6,25,26]. Peripheral blood mononuclear cells (PBMCs) were obtained by ficoll gradient and stored frozen in liquid nitrogen until use.

### 2.3. Healthy Donor (HD)

HD samples were obtained from written informed donors and collected by clinicians of the CHU Montpellier and collected and processed as patient’s samples.

### 2.4. Clinical Criteria for CMV-Seropositive Patients

Previous CMV infection was resolved as follows: IgG < 12 U/mL = >negative; IgG > 14 = >positive; IgM <18 = >negative; IgM > 22 = >positive; IgG +/IgM− = >Profile for an old infection; IgM + = >First infection or recent reactivation; IgG− IgM− = >never in contact. At the moment of sampling, none of patients presented symptoms of CMV reactivation; however, we do not test patients for current CMV reactivation.

### 2.5. Fluorescence-Activated Cell Sorting (FACS) Analysis and Antibodies

In brief, healthy donors and patient’s total PBMC were stained for surface markers with the following fluorochrome conjugated antibodies: CD19-BUV737, CD107a-BUV395, CD3-BV786, CD33-BV711 & CD14-BV650 (for AML study), CD15-BV650 & CD30-BV605 (for HL study), PD1-FITC, CD62L-PerCP-Cy5.5, CD57-PE-CF594, CD16-AF700 from Beckton Dickinson, CD69-PE from Beckman Coulter, CD45RO-VioGreen, CD7-VioBlue, CD56-PE-Vio770, CD45RA-APC-Vio770, and NKG2C-APC from Miltenyi Biotec. Cell viability was determined using DAPI (BD Biosciences). Cells were stained with the corresponding antibodies in FACS buffer (PBS, 2% FBS) on ice for 25–30 min and washed twice with the same FACS buffer and acquired on BD LSR-Fortessa instrument (Blue-Yellow/Green-Red-Violet-Ultraviolet) (BD Bioscience). Flow Cytometry Standard (FCS) files were analyzed using FlowJo software v10.6.1 (Becton, Dickinson and Company, Ashland, OR). The gating strategy for conventional flow cytometry and the determination of the “percentage” of selected cells in the figures is described in the Appendix A.

### 2.6. High Dimensional Reduction Analysis

To generate Uniform Manifold Approximation and Projection (UMAP) embedding of multi-parameter FACS, data from HD and patient per timepoint and difference in NK cell number between samples are normalized using FlowJo Downsample plugin (v3.1.0) in such a way that each sample contains an equal number of NK cells after downsampling, as this will allow equal representation of variation in NK cell subsets from multiple samples and normalized for difference in sample size. Finally, downsampled NK cell populations from different donors are merged into one single FCS file using sample concatenation function in FlowJo before clustering by UMAP plugin (v2.2) [27]. All processes were performed under FlowJo software v10.6.1.

### 2.7. Correlation Matrix Analysis

The correlationship between percentages of various NK cell and tumor cell populations was analyzed by Pearson correlation method with confidence interval (CI) = 95% and the resultant correlation matrix was visualized using the package corrplot (v0.84) in R environment (v3.6.1).

### 2.8. Statistical Analysis

Experimental figures and statistical analysis were performed using GraphPad Prism (v8.0). All statistical values are presented as * *p* < 0.05; ** *p* < 0.01; *** *p* < 0.001, and **** *p* < 0.0001. Mean values are expressed as mean plus or minus the standard error of the mean (SEM).

## 3. Results

### 3.1. Antitumor NK Cells in HL Patients are Identified by CD45RARO and CD107 Expression

To investigate how the chronic presence of tumor target cells and their subsequent clearance after treatment affect the NK cell population, we used a cohort of 21 patients from the Hematology Department, CHU Montpellier, France with Hodgkin Lymphoma (HL). We collected and analyzed blood samples at diagnosis and after treatment and compared them to those of healthy donors (HD). The treatment regimen for these patients varied in terms of the chemotherapy used and the combination with radiotherapy (Table 1). Since these patients received a variety of treatments, we could not subgroup them based on treatment type with very few individuals, and we analyzed the whole cohort. To analyze the NK cell population in these patients, we performed analysis based on both manual gating strategies and unsupervised UMAP mapping approach (Appendix A). NK cells are usually detected by CD56 expression. However, some myeloid subsets can express CD56 [28]. In agreement with previous results [29], more than 90% of CD56+ cells in our samples expressed CD7 (Appendix A). Hence, we used CD7 as a marker of the lymphoid lineage to identify bona fide NK cells and analyzed CD7+/CD56+ cells [28].

In general, HL patients show low numbers of tumor Reed Sternberg cells (RS), which are usually defined by CD15 and/or CD30 expression, in blood [30]. In agreement, we also found few CD15^+^/CD30^+^ cells in our cohort (Appendix A), although we cannot affirm that these cells are tumor cells. Treatment modified the percentage of CD15^+^/CD30^+^ cells, but the effect was highly variable, with patients showing increasing values while in others there was a decrease (Appendix A). This heterogeneity was also observed in B, T, and NK cells, although T cell numbers usually decreased (Appendix A). The percentage of T cells also showed a tendency to decrease, whereas the percentage of NK cells increased (Appendix A).

Next, we analyzed changes in NK phenotype after patient treatment using a manual gating strategy. CD56^bright^ cells represent an immature phenotype, whereas CD56^dim^CD16^+^ cells represent the most mature subpopulation [5,6,7]. Patients at diagnosis showed a tendency to decrease the percentage of mature NK cells compared to HD, and this correlated with an increase in the most immature CD56^bright^ subpopulation (Figure 1A). After treatment, the changes in these different NK cell subsets were statistically significant. This suggests a most immature state of the NK cell compartment after treatment, as previously observed in diffuse large B-cell lymphoma (DLBCL) and follicular lymphoma (FL) patients [7].

The antitumor NK cell population is recognized by expression of CD45RO (CD45RO cells) in general together with CD45RA (CD45RARO cells; [5,6,7]. At the time of diagnosis, hematological cancer patients show high levels of these cells together with CD45RO^+^ cells, generally leading to a decrease in the CD45RA^+^RO^−^ population (CD45RA cells) [5,6,7]. Patients in our cohort clearly showed this phenotype (Figure 1B). At the end of treatment, CD45RARO cells decreased. Taken together, these data suggest that elimination of target cells decreases NK cell activation status. The population that significantly increased was CD45RAdimRO^−^, which is immature and are normally CD56^bright^ [5,6]. This could partially explain the increase in immature NK cells (Figure 1A).

We next analyzed expression of several NK markers. During in vivo maturation, CD56^bright^ cells become CD56^dim^CD62L^+^CD57^−^ cells that produce perforin, while maintaining high IFN-γ production in response to cytokines [31,32]. On the other hand, CD56^dim^CD62L^−^CD57^+^ cells show lower responsiveness to cytokines and higher cytotoxic capacity [31,33]. CD69 is an activation marker [5,6,7]. NKG2C^+^ NK cells accumulate in cytomegalovirus (CMV)–seropositive human adults [9] and NKG2C could be a marker of memory NK cells [23]. The presence of CD107 on the cell surface ex vivo is a sign of in vivo degranulation of NK cells [5,6,7]. PD-1 is an immune checkpoint, which is absent on NK cells isolated from healthy donors but it is expressed on those from certain hematological cancer patients’ [34]. However, NK cells from patients show a large heterogeneity of PD-1 expression and only a few of them constitutively express PD-1 [7].

NK cells from patients did not significantly change expression of CD57, CD69, CD62L, and NKG2C compared to those from HD (Figure 2A). However, we observed a large heterogeneity, with some patients that largely increased the percentage of NKG2C- and CD69-expressing NK cells. Interestingly, in patients, there were increased numbers of NK cells expressing CD107a compared to HD. Our data demonstrated that the NK population in HL is affected by the presence of tumor cells as previously described in other hematologic cancers [5,6,7]. Surprisingly, certain patient’s NK cells expressed PD-1, suggesting that they can be inhibited by PD-L1 expressing cells.

After treatment, the proportion of NK cells expressing CD57 decreased (Figure 2A). This supports again an increase in the immature subset of the NK compartment. Percentages of CD62L, CD69, and NKG2C expressing cells did not change. On the other hand, PD-1 expression tended to decrease (Figure 2A). Hence, the continued renewal of NK cells might preclude higher levels of PD-1 expression on NK cells in some patients. In agreement with our previous observations [5,6,7], the decrease of tumor target cells after treatment diminished CD107 expression on NK cells.

To further investigate the antitumor NK cell population, we looked at the gain of tumor markers by trogocytosis [5,6,7]. RS tumor cells usually express CD15 and CD30 [35], but lack CD19 or CD20 expression [30]. Interestingly, the percentage of NK cells carrying CD19 did not change in patients before or after treatment (Figure 2B). In contrast, we observed that NK cells gained the CD15 and CD30 RS markers by trogocytosis. Differences were statistically significant when we analyzed cells that had trogocytosed on either CD15 or CD30 or both (Figure 2B). The percentage of NK cells that have trogocytosed dropped after treatment and became non-statistically significant. There were still some patients with NK cells carrying CD15 and/or CD30 after treatment. This could be related to the presence of remaining tumor cells, i.e., the tumor clearing is still uncompleted, or to NK cells that had previously trogocyted those antigens and were still alive. The degree of significance largely increased when patient 21 was taken out of the statistical analysis. In contrast to other patients, this patient showed a large increase in trogocytosis after treatment, and was the only one in the cohort that was HIV seropositive (Table 1). In summary, we found a contraction of the antitumor NK cell population after treatment, as we had previously observed in DLBCL and FL patients [7]. We next stratified patients regarding their treatment and analyzed their response (Appendix A). All groups equally responded to treatment regarding a decrease in CD45RARO population, CD107a expression, and trogocytosis. This shows that different treatments induced similar effects and supports the notion that a specific treatment does not lead to our observations.

### 3.2. UMAP (Uniform Manifold Approximation and Projection) Identifies New NK Cell Subsets

To more precisely characterize how the presence of target cells, i.e., at diagnosis, or their absence, i.e., after treatment, affect the NK cell population, we used the high dimensional reduction algorithm UMAP to generate unsupervised UMAP embed maps with 14 NK surface markers (Appendix A). As described in Figure 3A, we generated a UMAP plot for all patients at the 2 timepoints (*n* = 42) and the 4 healthy donors (HD, *n* = 4). Each sample contained 4000 NK cells. From this original map, we derived 3 daughter maps only showing the events from HD, patients at diagnosis, or after treatment (Figure 3A). We identified 7 clusters common for both HD and patients (1 to 7) and 4 clusters that exclusively appeared in patients (HL_1 to HL_3 and cluster trogocytosis). There were more samples coming from patients than from HD, hence the clusters could better represent patient’s clusters. However, Figure 3A showed that the vast majority of HD NK cells were included in the selected clusters. This means that patients still keep the HD clusters, meanwhile showing new clusters.

Cluster HL_1 was almost exclusive of patient 21, who was HIV seropositive. Interestingly, this cluster contained mainly adaptive-like NK cells (CD57^+^NKG2C^hi^) (Appendix A and Figure 3B and Figure 4C).

Only cluster 3 significantly decreased in patients (Figure 3B), whereas cluster 5 tended to increase. Cluster 3 mainly comprised CD56^dim^CD16^+^ subset with CD57^−^NKG2C^low^CD7^hi^ phenotype (Figure 4). Interestingly, cluster 3 decreased more in advanced stages of HL (Figure 3C). The specific HL_ clusters were present in some, but not all, patients (Figure 3B,D). This suggests that they are not representative of a “general” HL-associated NK population, but mainly of the status or the specific conditions of the patient.

To further interrogate the phenotype of these clusters, we plotted the heatmap of expression on each surface marker that constituted the UMAP embed map (Appendix A). Characterization of phenotypes for each cluster identified from UMAP (Figure 4) included all cells that belong to each cluster, regardless from which sample they are derived from. This means clusters 1–7 contain NK cells from both HD and patients, whereas cluster HL 1–3 and cluster_trogo are only exclusively from patients because HD do not contain such clusters. Clusters 1 to 4 represent mature CD56^dim^CD16^+^CD45RA^+^CD45RO^−^ populations (Figure 4). Clusters 5 to 7 represent immature CD56^bright^CD16^−/low^CD45RA^+^CD45RO^−^ populations, with cluster 7 containing the CD45RA^dim^ population specific of hematological cancer patients [5,6,7]. The HL_1 to 3 represent populations with CD56^dim^CD16^low^ that can be discriminated by the expression of two markers NKG2C and CD57. HL_1 expressed both, HL_2 expressed only CD57 and HL_3 did not express any of them. Finally, the antitumor population was found in a specific HL cluster that we called cluster trogocytosis: CD56^dim^CD16^+^CD45RA^low^CD45RO^+^CD30^+^CD15^+^CD107^+^PD-1^+^. After treatment, this cluster decreased when we removed the HIV+ patient 21 from the analysis (Appendix A). Additionally, clusters 5 and 7 also changed, with a significative increase in the 7, which contained the immature CD45RA^dim^ cells (Appendix A). Hence, the immature population increased after treatment, although the exact markers are patient specific. Finally, correlation analysis showed a negative relationship between cluster 3 and clusters HL_2/3 (Appendix A). The trogocytosis cluster positively correlated with CD15+/CD30+ cell counts.

### 3.3. Effect of CMV Infection on NK Subsets in HL Patients

Several lines of evidence suggest that viral infection, i.e., EBV, CMV, or HCV, influences NK repertoire [36,37,38,39]. In addition, a possible link between CMV infection and Hodgkin’s lymphoma has been suggested [40]. Therefore, we examined the impact of previous CMV infection on the NK cell compartment in our HL cohort, which we divided in CMV-negative (patients 2, 3, 6, 8, 10, 11, 12, and 22, *n* = 8) and CMV-positive (patients 4, 5, 14, 17, and 18, *n* = 5) groups (Table 1). None of the patients had symptoms of CMV reactivation at the moment of sampling. We used the same UMAP and analyzed CMV^+^ vs. CMV^−^ patients. Cluster 3 was significantly lower and clusters HL_2 and 3 were significantly higher in seropositive patients (Figure 5A). However, treatment did not differently change these clusters in both type of patients (Figure 5B). It is interesting to note that cluster 3 is mainly composed of CD57^−^NKG2C^+^ cells, whereas clusters HL_2/3 are NKG2C^−^. Hence, in HL, previous CMV infection does not seem to correlate with an increase in NKG2C^+^ populations.

We analyzed single markers by manual gating in both CMV serotypes (Appendix A). Treatment induced a higher increase of the CD45RA^dim^ subset in the CMV^−^ group, and more interestingly a higher decrease of the antitumor CD45RARO subset (Appendix A). It is tempting to speculate that CD45RARO cells are kept in CMV^+^ patients to fight against CMV-infected cells. In agreement with this idea, CMV^−^ serotype showed a decrease in PD-1, CD69, and CD107a after treatment that was absent in CMV^+^ serotype (Appendix A). Moreover, treatment decreased trogocytosis mainly in the CMV^−^ serotype (Appendix A). This analysis also validated that seronegative patients had larger numbers of NKG2C^+^ NK cells (Appendix A). In contrast, CMV serotype did not change the percentage of CD62L and CD57 positive cells. In summary, NK cell remodeling after anticancer treatment depends on CMV serotype and, remarkably, CMV^−^ patients underwent a more robust shift in the NK cell compartment following treatment. Finally, we could analyze 2 patients in relapse: patient 4 (CMV^+^) and patient 8 (CMV^−^). Appendix A showed that both patients equally recovered HL_ clusters after relapse.

### 3.4. NK Cells Subsets in AML Patients

We had previously identified the antitumor NK cell population in AML patients [5,6]. As previously described, the NK compartment is enriched in CD45RARO and CD45RO cells that have degranulated, as evidenced by CD107a, and have performed trogocytosis. We investigated here the NK phenotypes linked to the presence of target cells at the time of diagnosis or their absence at the end of treatment. Thus, we collected and analyzed blood samples from 6 patients both at diagnosis and ≥30 days after the end of the first chemotherapy treatment (post-induction therapy). For one patient, we got an additional sample after relapse, which occurred 6 months after the end of treatment. Table 2 describes patients’ stage and treatment, which involved different chemotherapies. In AML, the efficacy of treatment can be followed by the decrease in blood of myeloid CD33+ blasts, which can also express CD14 (Appendix A). CD33 is a myeloid marker commonly used for the diagnosis of AML, and CD14 can be expressed in some AML types [41]. After treatment, NK cells decreased in patients with previous high blast numbers and increased in those with low numbers. Blast reduction was also observed using UMAP embed maps (Appendix A). To unveil NK cell subsets affected by the presence of target cells, we generated a UMAP embed map from total samples in which 2900 representative NK cells from each sample were included. NK cell subpopulations in the final UMAP map were clustered based on their similarities in the level of expression of 12 surface markers (Appendix A and Figure 6A, left panel). To assess the NK cell phenotypes associated with different disease stages, we extracted from the map containing all samples either those events from samples taken at diagnosis or from samples taken after treatment. This way, two additional daughter maps were generated describing diagnosis-related and post-treatment-related NK cell (Figure 6A,B). We identified 6 clusters with notable differences in density (Figure 6B). To further verify the differences in those 6 clusters between diagnosis and after chemotherapy (CT) samples, the percentage of cells in each of these clusters were quantified for the 6 patients (Figure 6C). This revealed a decrease in cluster 1 and 2 and an increase in cluster 4 after treatment. Changes in other clusters were patient-dependent without a common pattern.

We next sought to unveil the phenotype linked to each cluster. Appendix A displayed heatmaps of the level of expression of each marker used to constitute the UMAP. Clusters 1 to 3 lacked immature CD56^bright^ cells, which were present in clusters 4 to 6 (Figure 7A). Accordingly, clusters 1 to 3 express CD57 cells whereas clusters 3 to 5 mostly expressed CD62L (Figure 7A).

Cluster 1 contained the antitumor CD45RO^+^ NK cells (Figure 7A), including the NK cells that have degranulated, i.e., CD107a^+^, but also the PD-1^+^ cells (Figure 7B). Moreover, this cluster contained the cells that have trogocytosed the tumor antigen CD33 (Figure 7B). The percentage of CD33^+^ cells reached even 15% in some patients and strongly diminished when the target population decreased after treatment (Figure 7C). Although some NK cell subsets could contain CD33 mRNA and express CD33 [42], we believe that we observed CD33 trogocytosis, because healthy donors’ NK cells express much less CD33. Moreover, after treatment, when tumor targets are sparser, CD33 decreases in patients’ NK cells. Finally, only cells that have degranulated express CD33. In summary, CD33 expression is closely related to a specific NK cell subset with antitumor properties.

CD14 trogocytosis was detected although at a lesser extent, likely due to the fact that not all tumor cells express CD14. CD14 expression is related to the AML type. M0 and M2 blasts do not express CD14, M4 always express it, and M1 can express it [41]. Our cohort has patients of these four types (Table 2). M0 and M2 patients showed the lowest CD14 trogocytosis, whereas the highest was shown by a M4 patient (Figure 7C). This suggested that NK cells acquired CD14 by trogocytosis on tumor AML cells as they gained CD33. In summary, cluster 1 contains the bona fide antitumor NK population [5,6,7] and cluster 2 are mature CD57^+^ NK cells that decreased in percentage after treatment. In contrast, a similar population expressing CD62L did not decrease. Clusters 4 to 5 include immature NK cells, which arise when the target cells disappear.

The amount of AML blasts (CD33^+^CD14^−^) positively correlated with cluster 2 and negatively with cluster 4 in total samples or samples at diagnosis (Appendix A). After treatment, AML blasts correlated negatively with clusters 1 and 5 and positively with cluster 3.

We next stratified patients regarding their treatment and analyzed trogocytosis before and after treatment (Appendix A). As in HL, all groups equally responded to treatment regarding trogocytosis, this supports again the notion that a specific treatment does not lead to our observations.

Finally, we obtained samples from the AML patient 2 after relapse. We generated a set of specific clusters (Appendix A) and observed that clusters 1 and 2 largely disappeared after treatment and reappeared in relapse (Appendix A). These clusters were largely enriched in the antitumor CD45RARO population, which contained PD-1- and CD69-expressing cells and had also gained by trogocytosis the tumor antigen CD33 (Appendix A). This clearly showed that the presence of target cells generated the presence of these clusters.

## 4. Discussion

We have used here a non-supervised UMAP analysis of multiparametric FACS data to unveil new classical NK cell (cNK) subpopulations or clusters in HL and AML patients. We first verified whether HL patients possess similar NK subsets at diagnosis (Figure 1 and Figure 2) than patients with other hematological malignancies such as AML [5,6,7]. For this purpose, we focused on bona fide peripheral blood NK cells that we identified by CD56 and CD7 expression [28,29]. We next used UMAP to unveil clusters that can be associated to disease before and/or after treatment. We could not compare clusters in samples from both diseases between them because in order to study trogocytosis, we used CD33 and CD14 in AML, and CD15 and CD30 in HL. Nonetheless, in both diseases, NK subsets were deeply affected by the presence of tumor target cells.

Every individual has one’s own NK cell repertoire, which changes during development and is differentially distributed in different tissues. Moreover, multiple environmental factors can shape the NK cell compartment and generate new subsets [1]. This makes it extremely difficult to identify disease-associated NK subpopulations shared by multiple patients. Although we revealed some disease-specific populations, they were not present in all patients. This could be due to the fact that each patient has his/her own NK cell subsets due to different genetic backgrounds that influence the expression patterns of multiple receptors. Additionally, environmental factors, such as the history of prior infections to which each individual is exposed, can differentially impact the NK cell populations [43].

We have also shown that treatment largely modified the NK compartment. This could be due to the direct effect of treatment, i.e., chemotherapy, on NK cells or to the presence/absence of target cells. The lifespan of cNK cells is, on average, of 1 week in blood, implying a quick renewal [8,44], Since treatment was stopped at least 30 days before analysis, we favor that the elimination of tumor target cells, rather than treatment itself, was the inducer of changes in NK populations. Moreover, all treatments in both diseases similarly affect NK cells (Appendix A). Therefore, our data provide evidence of a quick response and adaptation of the NK cell compartment to target absence: a large increase in immature population, whereas antitumor NK cells disappeared. However, there is a third possible effect of chemotherapy that we cannot exclude and that can affect NK cell subsets at relatively long time. Chemotherapy could affect the generation and differentiation of NK cell precursors in the bone marrow, and because NK cell development can take more than one month, chemotherapy effects might persist even more than 30 days after the treatment ended.

The cytotoxic potential of NK cells was severely diminished at HL diagnosis, independently of the clinical stage. This deficiency is normalized 6 weeks after completion of treatment [45]. Failure to respond or evidence of early relapse do not improve NK function. Hence, NK activity could be a biomarker of clinical response and prognosis [45]. In agreement with this, and by manual gating, we identified that the antitumor NK subpopulation disappeared after treatment and that the NK population was normalized. In addition, using unsupervised analysis, we were able to identify new NK subsets that had not previously been described, e.g., cluster 3, which contained CD57^−^NKG2C^+^ NK cells, or cluster trogocytosis, which gathered NK cells that had degranulated and performed trogocytosis. Hence, we revealed new NK cell subpopulations in this pathological context.

Similarly, in AML patients, NK activity is suppressed at diagnosis, restored at remission, and suppressed again at relapse. Hence, the anti-tumor activity of NK cells inversely correlates to disease advancement [46]. Our unsupervised analysis revealed clusters clearly associated to diagnosis, i.e., cluster 1. This cluster is enriched in: (i) CD45RARO cells, which we have shown to contain antitumor cells, (ii) cells that have degranulated (roughly 50%), and (iii) cells that have performed trogocytosis (100% for CD33). In summary, these cells present all the characteristics of cells that have interacted with target cells and we consider this cluster the antitumor NK cell cluster, similar to the cluster trogocytosis in HL patients. Both antitumor clusters sharply decreased after treatment. Therefore, these data together with our previous results [5,6,7] support the notion that studying NK cell subsets could help to develop new methods on prognostic and/or diagnostic in cancer. A recent study also suggested that this would be the case in solid cancer patients [47].

Analysis of other human NK populations is more challenging [1]; for example, we did not investigate tissue-specific NK cells that will require biopsies. Hence, it is likely that some disease-associated populations escaped from our analysis. In addition, we have used an important but limited panel of NK cells markers. Extending this panel in the future will likely help to identify new subpopulations. In any case, we were unable to identify tumor-associated NK cell populations that remained after treatment. We confirmed the presence of an antitumor NK population, but it disappeared or drastically decreased after target clearing. In hematological cancers, cNK cells should be in contact with tumor cells, as these reside in the same organs that NK cells and their precursors, i.e., bone marrow and/or secondary lymphoid organs. This co-localization suggests a direct tumor effect on NK cell activation, maturation, and/or development. For example, AML blasts infiltrate bone marrow and reduce NK cell development [48,49]. Remarkably, solid tumor patients also possess strong NK cell dysfunction [20], indicating that invasion of NK cell niches is not a prerequisite for cancer-related NK dysfunction. In our study, the tumor/NK cell target interaction was revealed by the presence of NK cells that have trogocytosed tumor antigens (Figure 2 and Figure 5). Therefore, although we have not investigated a tumor infiltrating population such as that described in solid tumors, we are confident that we have investigated cNK cells that were in contact with cancer cells.

Previous infection with certain viruses, e.g., CMV, sculpts the NK population [9,10]. In cancer patients, the appearance of specific subsets in blood dictated by the presence of the tumor is not clearly defined [16], although it is clear that the NK lineage is dysfunctional [18,19,20]. Surprisingly, CMV-infected HL patients lost the CMV-related NK population, i.e., CD57+NKG2C+ cells. This suggests that, faced with a new target challenge, new NK populations overshadow the “old” populations to respond to the new challenge. Interestingly, CMV [8,16] and EBV reactivation [36] favors patient survival after a cancer-related allogeneic transplantation. Hence, acute viral infection can improve NK cell-mediated antitumor function. However, in our study, patients did not show symptoms of CMV reactivation and our cohort is too small to reach any conclusion regarding the survival of previously infected CMV patients.

## 5. Conclusions

The NK cell lineage is heterogeneous with multiple subsets that may have specific activity against different attacks. We show here that cancer induces a broad change in this lineage that allows identification of cancer patients and could be used to follow the effectivity of cancer therapies such as monoclonal antibody treatment. However, we were unable to identify cancer-sculpted NK cell populations or “memory-like” subsets with the NK cell markers used that are present all patients.

## Figures and Tables

**Figure 1 vaccines-08-00727-f001:**
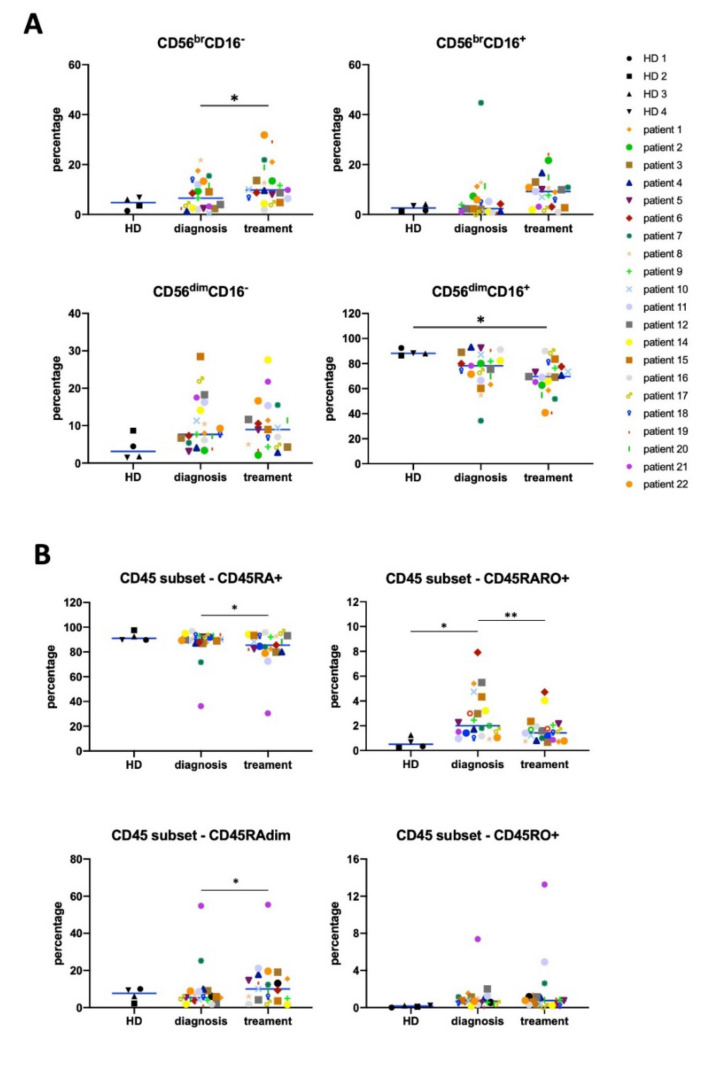
Hodgkin lymphoma (HL) natural killer (NK) cell subsets change after treatment. NK cell phenotypes were analyzed regarding expression of CD56 and CD16 (**A**), or CD45RA and CD45R0 (**B**), giving four different populations for each panel based on the expression of these markers. Statistical significance between HD (*n* = 4) versus patients (*n* = 21) were determined by one-way analysis of variance (ANOVA) (Dunnett’s correction). Statistical significance between “diagnosis” and “after treatment” for each patient was compared by the paired t-test, *n* = 21, * *p* ≤ 0.05, ** *p* ≤ 0.01,

**Figure 2 vaccines-08-00727-f002:**
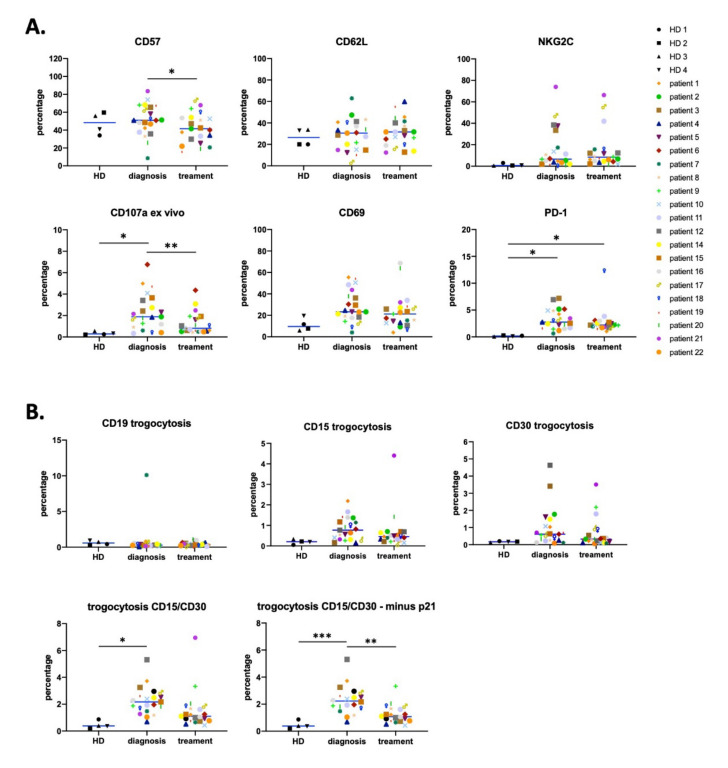
NK cell phenotype analyzed by manual gating. (**A**) Percentage of cells expressing PD-1, CD69, NKG2C, CD57, CD62L, and CD107a (degranulation ex vivo) on NK cells from healthy donors (HD) and from HL patients at diagnosis and after treatment. (**B**) Trogocytosis on tumor surface markers, i.e., CD19 (negative control), CD15, and CD30 (Reed Sternberg cells (RS) markers) by NK cell before and after treatment. In the bottom left graph, we represented NK cells expressing CD15 and/or CD30. The right graph represents the same but excluding patient 21, who was HIV+. Statistical significance between HD versus patients were determined by one-way ANOVA (Dunnett’s correction), “diagnosis” versus “after treatment” for each patient was compared by paired t-test, *n* = 21, * *p* ≤ 0.05, ** *p* ≤ 0.01, and *** *p* ≤ 0.001.

**Figure 3 vaccines-08-00727-f003:**
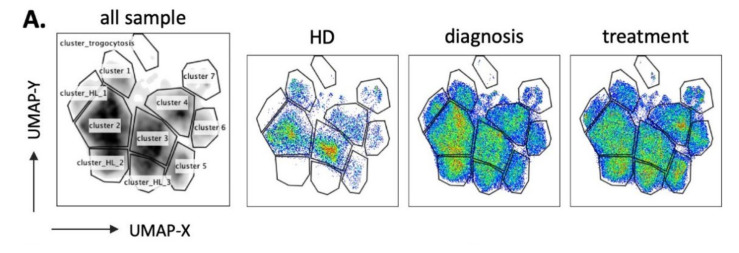
Identification of NK cell clusters in HL patients via Uniform Manifold Approximation and Projection (UMAP). Four thousand representative NK cells from HD (*n* = 4) and HL patients (*n* = 21, 2 timepoints per patient (diagnosis and treatment); total *n* = 46) were combined to generate an UMAP embed from expression level of 14 surface markers. (**A**) Original UMAP plot of all samples and daughter maps from HD, HL patients at diagnosis, and HL patients after treatment. (**B**) Each cluster was assigned an annotation and subsequently quantified in each HD and patient. (**C**) Proportion of cluster 3 plotted against HD and various HL stages. (**D**) Cluster quantification as shown in B with patient ID labelled. Statistical significance determined by unpaired t-test, *n* = 21, * *p* ≤ 0.05, ** *p* ≤ 0.01.

**Figure 4 vaccines-08-00727-f004:**
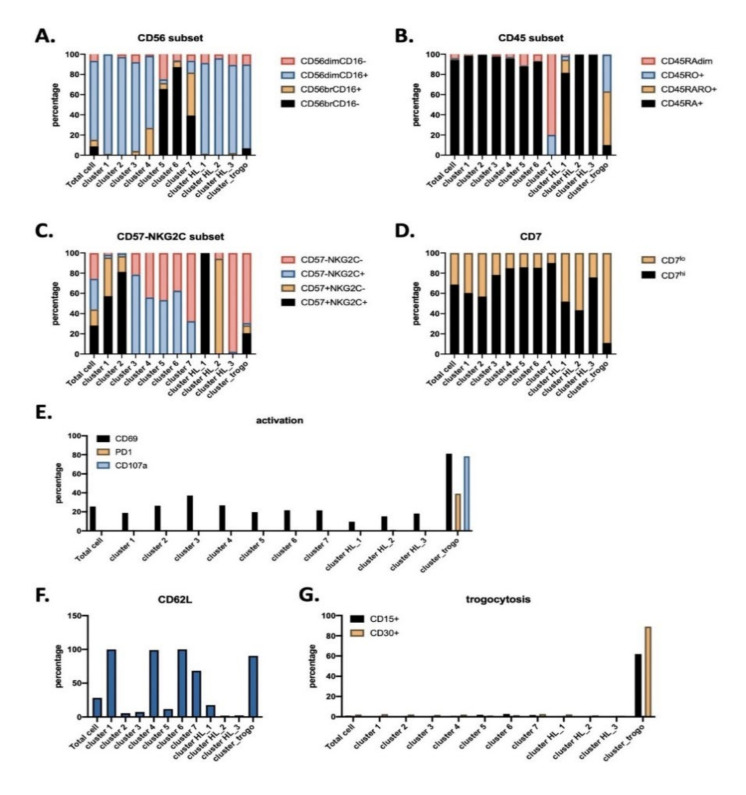
Phenotype of NK cell clusters identified via UMAP. Graphs represent for each individual cluster the percentage of the different NK cell subset. **A–G**) characterization of NK phenotype regarding to: CD56 subsets (**A**), CD45 subsets (**B**), CD57 vs. NKG2C (**C**), CD7hi (immature) vs. CD7lo (mature) (**D**), activation markers (i.e, CD69, PD1, CD107a ex vivo) (**E**), CD62L (**F**), and trogocytosis markers (**G**).

**Figure 5 vaccines-08-00727-f005:**
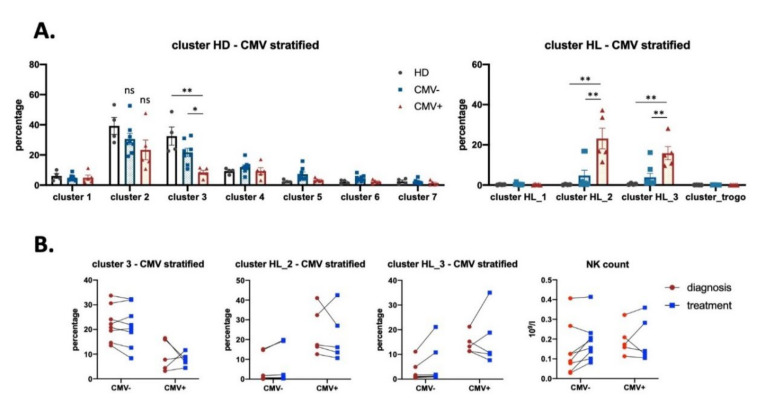
Analysis of NK cells from HL patients stratified by CMV serotypes. Graphs represent the percentage of NK cells in each cluster in HD, CMV^−^ (*n* = 8) and CMV^+^ (*n* = 5) patients (based on CMV serotype, although in the absence of CMV reactivation). (**A**) NK cluster quantification between HD and both CMV−/+ groups. (**B**) Change in NK cluster proportion before and after treatment in the 2 CMV groups. * *p* < 0.05, ** *p* < 0.01.

**Figure 6 vaccines-08-00727-f006:**
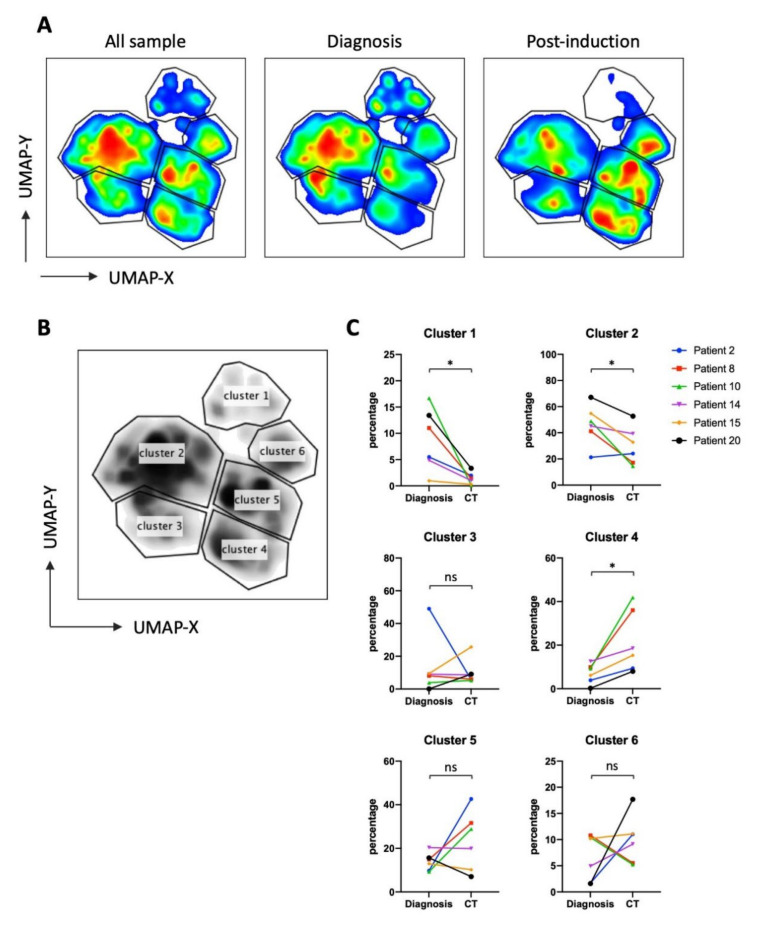
UMAP maps of NK cells in AML patients. We obtained 2 samples for each patient: at diagnosis and at least 30 days after finishing the first chemotherapy treatment (post-induction), which are described in Table 2. We used 2900 representative NK cells for each sample to pool and subsequently map on UMAP plot based on expression level of 12 surface markers used in the multiparameter flow cytometry panel. (**A**) UMAP plots of total (**left**) or diagnosis (**middle**) and after treatment (**right**) samples are shown. (**B**) Identification and annotation of NK cell clusters specifically enriched in either at diagnosis or at post-induction. (**C**) percentage of NK cell clusters (cluster 1–6) at different timepoints: diagnosis and post-induction (CT). Significance was determined by paired t-test between diagnosis versus treatment, with *n* = 6, * *p* ≤ 0.05.

**Figure 7 vaccines-08-00727-f007:**
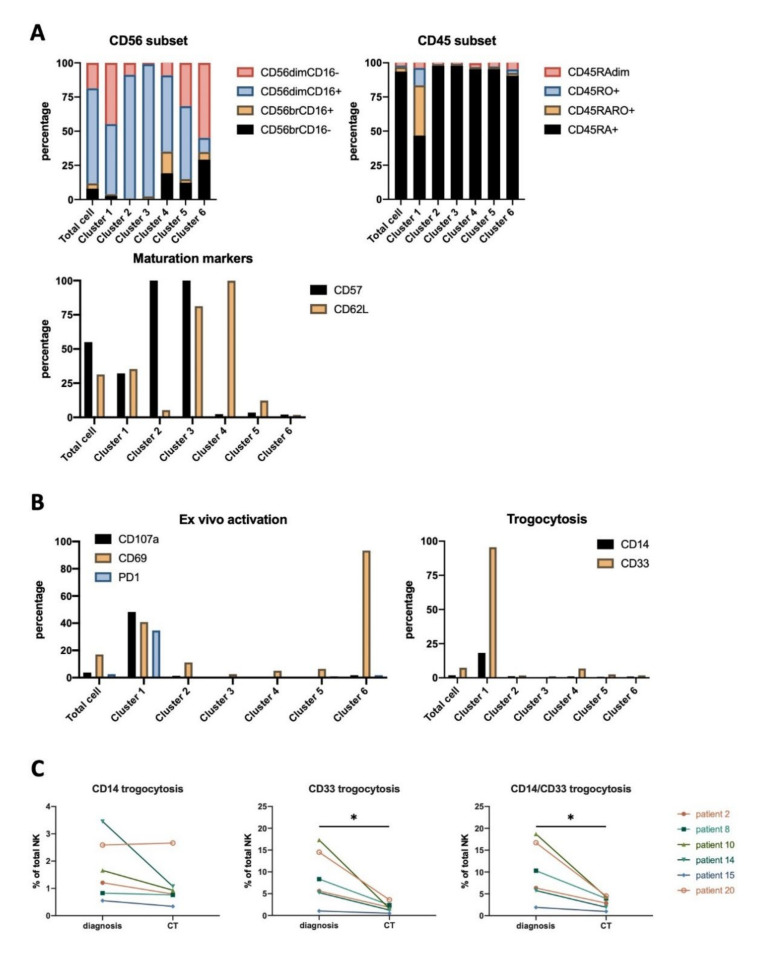
Phenotyping of NK cell clusters identified from UMAP maps in AML patients. Graphs represent for each individual cluster the percentage of the different NK cell subset. (**A**) Percentage of NK cells that express different levels of CD56 and CD16 (**left**), CD45RA and CD45RO (**right**), or CD57 and CD62L (**below**). (**B**) Percentage of cells expressing ex vivo CD107a, CD69, PD-1 (**left**) or that have performed trogocytosis on CD33 and CD14 (**right**). (**C**) Percentage of NK cells that performed trogocytosis of myelo-monocyte markers CD14 and/or CD33 at diagnosis and after therapy. * *p* ≤ 0.05

**Table 1 vaccines-08-00727-t001:** Clinical characteristics of the HL cohort. The table describes the patients’ subtype according to the Ann Arbor scoring system, the treatment, and the outcome. The cohort of 22 patients is composed of 12 men (55%) and 10 women (45%). Patients were between 19 and 69 years old.

Patient Number	Stage at Diagnosis (Ann Arbor)	First Treatment	Response (Lugano)	Relapse	First Treatmenton Relapse	Response at the End of Treatment	Second Treatment on Relapse	Last Medical Status	CMV/HIV
1	2	ABVD x3 + radiotherapy	CR	No	-	-	-	CR	na/−
2	3	BEACOPP x6 + AVD x2	CR	No	-	-	-	CR	−/−
3	1	ABVD x2 + R-DHAC x4 + BEAM + autograft	CR	No	-	-	-	CR	−/−
4	4	PVABx6	PR	Yes	DHAC x6	Progression	Brentuximab + Bendamustine x6	CR	+/−
5	4	BVAPx6	CR	No	-	-	-	CR	+/−
6	3	BEACOPP x2 + ABVDx4	PR	No	-	-	-	PR	−/−
7	4	EACOPP x6	CR	No	-	-	-	CR	na/−
8	4	PVAB x4	CR	Yes	R-DHAC x4 + autograft	Progression	R-CHOP x4	CR	−/−
9	2	ABVD x3 + radiotherapy	CR	No	-	-	-	CR	na/−
10	3	BEACOPP x2 + ABVD-MP x4	CR	No	-	-	-	CR	−/−
11	1	ABVD x8	CR	No	-	-	-	CR	−/−
12	2	ABVD x4 + radiotherapy	CR	No	-	-	-	CR	−/−
13	3	ABVD-MP x6	CR	Yes	DHACx4	Progression	Brentuximab + bendamustine	CR	na/−
14	4	PVAB x6	CR	No	-	-	-	CR	+/−
15	2	ABVD x4 + radiotherapy	CR	No	-	-	-	CR	na/−
16	2	ABVD x6 + radiotherapy	PR	No	-	-	-	PR	na/−
17	2	ABVD x3 + radiotherapy	CR	No	-	-	-	CR	+/−
18	2	ABVD x4 + radiotherapy	CR	No	-	-	-	CR	+/−
19	3	PVABx6	CR	Yes	Nivolumab x13	CR	-	CR	na/−
20	1	ABVD x3 + radiotherapy	CR	No	-	-	-	CR	na/−
21	4	BEACOPP x2 + ABVD x4	CR	No				CR	na/+
22	4	BEACOPP x2 + ABVD x4	CR	No	-	-	-	CR	−/−

List of abbreviations: Complete Response to treatment (CR), Partial Response (PR). Treatments: BEACOPP = Bleomycin + Etoposide + Adriamycin + Cyclophosphamide + Vincristine + Prednisone + Procarbazine; EACOPP = BEACOPP without bleomycin; ABVD = Doxorubicin + Bleomycin + Vinblastine + Etoposide + Cytarabine + Melphalan; R-DHAC = Rituximab + Carboplatine + Cytarabine + Dexamethasone + Dacarbazine; PVA = Prednisone + Vinblastine + Doxorubicin; PVAB = PVA+ Bendamustine; MP = Melphalan + Prednisone ; BEAM = Carmustine; R-CHOP = Rituximab + Cyclophosphamide + Doxorubicine + Vincristine + Prednisone.

**Table 2 vaccines-08-00727-t002:** Clinical characteristics of the acute myeloid leukemia (AML) cohort. Patients’ subtype according to the French-American-British (FAB) classification, treatment, and outcome of the AML cohort use in this study. Cohort of 6 patients is composed of 5 men (83%) and 1 woman (17%). Patients were between 25 and 59 years old. List of abbreviations: Complete Response to treatment (CR), High dose (HD), Medium Dose (MD); AMHAC = Amsacrine + Cytarabine.

Patient Number	AML Subtype	Treatment Induction	Consolidation	Response	Allograft	Relapse	Treatment on Relapse	Last Medical Status
2	M4	1st induction: Idarubicine + cytarabine 2nd cytarabine-HD	cytarabine HD	RC	Yes	6 months after graft	AMHAC + consolidation cytarabine-HD	deceased (myocard infarctus)
8	M2	1st induction: Idarubicine + cytarabine	cytarabine HD	RC	Yes	No	-	CR
10	M1	1st induction: daunorubicine + cytarabine	cytarabine HD	RC	Yes	1 month after graft	-	Deceased (AML relapse + GvH D)
14	M4	1st induction: Idarubicine + cytarabine	cytarabine HD	RC	Yes	2, 5 years after graft	gemtuzumab ozogamicin + mitoxantrone + allograft	CR
15	M0	1st induction: idarubicine + cytarabine. 2nd: cytarabine HD	cytarabine HD	RC	Yes	No	-	Deceased (GvH disease and infections)
20	M1	1st induction: cytarabine + idarubicine + azacitidine	-	RC	No	before graft	azaticidine + sorafenib	Deceased

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
