# Peer review of "Dissecting the NK Cell Population in Hematological Cancers Confirms the Presence of Tumor Cells and Their Impact on NK Population Function"

_vaccines, 2020, doi:10.3390/vaccines8040727_

Round 1
Reviewer 1 Report
Manuscript by Vo et. al. characterizes various subpopulations of NK cells in heath vs hematological cancer patients. Overall, study is interesting and well throughout. Manuscript is well written with some minor corrections below. I recommend publication with revision to address following points.
Line 164.168 and elsewhere: authors should use replaced % with percentage in the text, throughout.
Fig. 1: Author needs to show statistical significance between all groups of data. I also see marker description for two subsets only (immature vs mature), if other gates have no useful meaning they should be moved to supplementary figure. Figure legends need to add bit more of description of data shown.
Authors should also show dot plots of their gating strategy with each figure.
Fig. 3 Are there more clusters in patients, because there are more cells available for analysis compared to HD.
Authors need to elaborate Method section of UMAP clustering analysis. From result it was unclear whether samples had been downsized.
Fig. 4 Is this represent only cells from patients
Authors also mentioned they were investigating biological effects of therapeutic antibodies on NK population, please make this clear in results?
Reviewer 2 Report
This is an informative paper describing the results of multiparameter flow cytometric analyses on NK cell subsets in patients of hematological malignancies before and after chemotherapy. Data presented are new but relatively preliminary with small number of cases, and the authors should be more reserving in interpreting the effect of tumor cell burden on NK cell phenotypes as some data are inconsistent with their conclusions. However, the paper is worth publishing provided that the authors can successfully deal with the below major issues and go through modifications of the text especially on their peculiar English usage.
Major concerns:
As the authors themselves admit in lines 417-418, the observed changes in NK cell phenotypes could be caused by chemotherapy itself either directly through its effect on NK cells or, although the authors fail to mention, through its effect on the generation and differentiation of NK cell precursors in the bone marrow. The latter effect might persist even more than 30 days after the treatment ended.
The authors' presentation of data on trogocytosis is very interesting and important, but is inconsistent with their description of "tumor cell clearance" after treatment. In fact, in Figure 2, NK cells still show high levels of CD15 or CD30 trogocytosis after treatment in comparison with those in healthy donors, which is negligible. If tumor cells were cleared after treatment as the authors describe, how did NK cells acquire these tumor cell markers through trogocytosis after complete response (as stated in Table 1)? If NK cells are renewed at more than 30 days after the end of treatment as the authors describe in lines 419-421, the observed acquisition of CD15/CD30 must indicate the persistence of tumor burden after treatment. The authors must discuss more carefully how they have defined tumor clearance.
In Figure 6, what does "post-induction" mean in panel A? There is no description about "induction" in the text. In panel C, what does CT indicate? Is this after treatment?
Numerous minor points:
Throughout the text, NK, NK cells, and "NKs" are used inconsistently. "NK are" in line 13 does not make sense and this must be NK cells are. Similarly, NKs have in line 45 and NK have in line 64 are both used.
Line 14, ADCC is antibody-dependent cell-mediated (or cellular) cytotoxicity, and is not generated but mediated by mAbs.
Lines 25 and 26, the same NK populations as (not that) those found at diagnosis (not diagnostic).
Line 91, did all donors go through some kind of surgery? What surgery did they have?
Line 102, "patient CMV seropositive" must be CMV-seropositive patients.
Lines 106-107, what do the authors mean by "patients were not properly tested to exclude this"? Did the authors perform improper tests?
Line 115, what is antibodies cocktail? Do the authors mean a cocktail of antibodies or antibody cocktails?
Lines 127-128, what is the result correlation matrix? The resultant correlation matrix?
Line 139, do the authors mean treatment regimen by "treatment regime"?
Line 140, table 1 must be capitalized.
Line 170, it is unclear which changes were statistically significant.
Line 185, as discussed above, if target cells were eliminated, where did NK cells acquire CD15 and CD30 by trogocytosis?
Line 198, "statistically change" does not make sense. Do the authors mean significantly?
Line 203, certain patients' NK cells as this was observed in multiple patients.
Line 218, what does "larger PD-1 expression" mean? The expression of a larger PD-1 molecule? Do the authors mean higher levels of PD-1 expression?
Lines 229 and 245, was patient 21 HIV-1-positive by PCR or just seropositive?
Line 373, "healthy donors NK cells" must be healthy donors' NK cells or NK cells from healthy donors.
Lines 401-402, "similar NK subsets at diagnosis than those with other hematological malignancies" does not make sense.
Line 408, "his" here must be one's.
Lines 414 and 415, is "the record of prior infection" an environmental factor? Do the authors mean history of prior infection?
Round 2
Reviewer 2 Report
The authors have adequately responded to this reviewer's comments and have modified the text accordingly.